# RE-EXAMINING LEARNING LINEAR FUNCTIONS IN CONTEXT

## ABSTRACT

In context learning (ICL) is an attractive method of solving a wide range of problems. Inspired by Garg et al. (2022), we look closely at ICL in a variety of train and test settings for several transformer models of different sizes trained from scratch. Our study complements prior work by pointing out several systematic failures of these models to generalize to data not in the training distribution, thereby showing some limitations of ICL. We find that models adopt a strategy for this task that is very different from standard solutions.

## 1 INTRODUCTION

In-context learning (ICL) Brown et al. (2020) promises to make interacting with LLMs easy and accessible. ICL enables the model to learn a task from a prompt with instructions and a few examples at inference time, without any adjustment of the model's parameters from pretraining. While there have been theoretical reconstructions of ICL, there have been few studies on exactly how ICL works in practice. ICL depends on a model's pretraining; so doing an in depth analysis of this feature of LLMs is difficult. Hence, most of analysis done on how ICL works are done on small models and simple tasks. Garg et al. (2022) makes the problem mathematically precise: the model learns a task/function given in-context examples at inference time in a next-token-prediction format Brown et al. (2020); given a prompt containing a task input-output examples $(x_1, f(x_1), .., x_n, ?)$, the model is asked to generate a value approximating $f(x_n)$.

Inspired by Garg et al. (2022), we investigated whether smaller LLMs with transformer architectures ICL the class $\mathcal{L}$ of linear functions. While Garg et al. (2022) answer "yes", we provide a more nuanced answer based on a deeper analysis. We have studied the 1 dimensional case with functions for over 30 models, from transformer architectures with 1 attention head (AH) and 1 MLP layer up 12 MLP layers and 8 AH. We also studied small attention-only models Olsson et al. (2022). Since we are interested in whether transformer models can ICL and if so how, even small transformer models are relevant, indeed essential since such an investigation requires training from scratch. Our main findings are these.

1. Several recent papers claim that Transformer based models trained from scratch can through ICL implement algorithms like linear and ridge regression or Newton's method. By shifting sampling from different training and test distributions of both functions $f$ and values $x_i$, we show that the models we tested do not do this and fail to generalize or to provide robust predictions beyond their training data. In particular, all our transformer models failed to ICL the concept of a strictly increasing or strictly decreasing linear function, even over larger intervals in $\mathbb{R}$. We trained transformers on different distributions various Gaussian, Bimodal and Uniform distributions.

2. Our experiments show that all our models on all training distributions (though training with uniform distributions makes this particularly clear) have 'boundary values' $(B, -B)$ for prompts $x_i$; when $f(x_i) > B$ or $< -B$, model performance degrades substantially. We argue boundary values are crucial to understanding ICL.

3. All our transformer models solve the task of ICL linear function by learning a projection from "nearby" sequences of points in the training data; In Section 5 we model mathematically what we think the models do. The projection depends upon the training distribution.

## 2 BACKGROUND

Neyshabur et al. (2017), Villa et al. (2013) define learnability in statistical learning theory via the notion of *uniform consistency*. Let $\mu$ be a distribution over $\mathcal{H}$ and $\mu_n$ the update of $\mu$ after $n$ training samples $z_i = (x_i, y_i)$. Let $A_{z_n}$ be an algorithm for picking out a hypothesis from $\mathcal{H}$ based on $n$ training samples. $inf_{\mathcal{H}}$ is the hypothesis in $\mathcal{H}$ with the lowest possible error (Shalev-Shwartz et al., 2010; Kawaguchi et al., 2017).

**Definition 1** *An algorithm $A$ on a hypothesis space $\mathcal{H}$ is uniformly consistent if and only if*
$\forall \epsilon > 0 \, lim_{n \to \infty} sup_{\mu}$
$$\mu_n(\{z_n : \mathbb{E}_{\mu}(\{A_{z_n} - inf_{\mathcal{H}}\mathbb{E}_{\mu} > \epsilon\}) = 0$$

In our example, the best hypothesis $inf_{\mathcal{H}}$ is a prediction $\hat{f}$ of some target function $f$. The best hypothesis is when $\hat{f} = f$ with $f$, which yields 0 expected error. There is of course an algorithm that gives exactly the target function, linear interpolation, given two data points. Moreover linear regression is an algorithm that converges to the target function on any data set in our set up.

**Definition 2** *A class of hypotheses $\mathcal{H}$ is* uniformly learnable *just in case there exists a uniformly consistent algorithm for $\mathcal{H}$.*

The class of linear functions $\mathcal{L}$ is clearly uniformly learnable. What is left open here is the choice of distribution of the data both for train and test and the sampling method (since our class is uncountably large). Garg et al. (2022) take a definition of learning where average expected error goes to 0 when data in train and test are sampled both from the same normal distribution. However, a class of mathematical functions like $\mathcal{L}$ does not in any way depend on a particular distribution or sampling. And so we would expect that if the model has ICL $\mathcal{L}$, it has found an algorithm such that $\hat{f} = f$ given a test set of linear functions and points not in its training distribution. In such a case the model will ICL with different distributions. This is what we investigate below.

## 3 RELATED WORK

Since Brown et al. (2020) introduced ICL, there has been considerable research indicating that ICL is possible because of a sort of gradient "ascent" Akyürek et al. (2022); Von Oswald et al. (2023). Dong et al. (2022) provides an important survey of successes and challenges in ICL and that so far, only simple problems for ICL have been analyzed, eg the case of linear or simple Boolean functions.

Garg et al. (2022) offered an important advance showing that a Transformer trained from scratch (GPT-2 with an embedding size of 256) performed in-context learning of n-dimensional linear functions given identical train and test distributions $N(0, 1)$.

Further research then offered several theoretical reconstructions for how ICL for linear functions might work in Transformers. Von Oswald et al. (2023); Ahn et al. (2023); Mahankali et al. (2023) provided a construction to show transformers ICL from their doing gradient descent during ICL. Fu et al. (2023) showed that Transformers could ICL in virtue of using higher-order optimization techniques. Xie et al. (2021); Wu et al. (2023); Zhang et al. (2023); Panwar et al. (2023) argued that ICL follows from Bayesian principles. Bai et al. (2024) show that transformers can under certain assumptions implement many algorithms with near-optimal predictive power on various in-context data distributions. Given Pérez et al. (2021)'s result that full transformers with linear attention are Turing complete, however, these theoretical demonstrations are perhaps not surprising.

Xie et al. (2021); Zhang et al. (2024) show that when we shift training and inference distributions ICL performance degrades. Thus, this work is closer to our own as is Giannou et al. (2024). However, Giannou et al. (2024); Zhang et al. (2024) make important modifications to transformer architectures Giannou et al. (2024); Zhang et al. (2024) work with linear attention, whereas we look at attention layers as they actually are used with softmax. In addition, Zhang et al. (2024) uses a new kind of optimization or training with gradients and a special fixed initial point. This means that their architecture and training are quite different from what normally happens with transformers; they are interested in getting a revised transformer-like model to learn linear functions, while we want to find out whether transformers as they actually are learn linear functions or something else. As we

detail below, the results for the architectures of Zhang et al. (2024); Giannou et al. (2024) are quite different from those we have for actual transformers. In addition unlike either of these papers, we show that prompts that are too long induce chaotic behavior.

Unlike this prior research, we examine how ICL works in practice under different training and testing distributions in order to establish what transformers *actually* do in ICL 1 dimensional linear functions, whereas most prior research has concentrated on transformer models *can* or *could* do on this task. Even for this simplest case, we show transformers ICL in a different way from any of these proposed methods.

Bhattamishra et al. (2023) trained small GPT-2 models from scratch to show that Transformers can ICL simple boolean functions, while their performance deteriorates on more complex tasks. Wu et al. (2023) studied ICL by pretraining a linearly parameterized single-layer linear attention model for linear regression with a Gaussian prior proving that the pretrained model closely matches the Bayes optimal algorithm. Raventós et al. (2024) investigated whether models with ICL can solve new tasks very different from those seen during pretraining.

Olsson et al. (2022) offer an in depth analysis of ICL across tasks using a general evaluation measure on prompt length. They propose that a learned copying and comparison mechanism known as an *induction head* is at the heart of ICL.

## 4 EXPERIMENTS

In this section, we show that: (i) models do not implement linear regression; (ii) this performance holds across different types of distributions; (iii) these distributions all show the presence of boundary values beyond which the models do not perform well; (iv) models with attention layers (AL) (models with at least two AL only or 1 AL+MLP layer) are needed to give an ICL effect (v) ordering and restricting the order of prompts can improve performance. In the last subsection, we put all of these observations together.

We trained several small decoder only transformer models from scratch to perform in-context learning of linear functions.[1] We set the number of layers (L) from 1 to 6, and attention heads (AH) from 1 to 4. We also trained a 9L6AH model and the 12L8AH GPT2 with an embedding size of 256. The task of the model is to predict the next value for $f(x_i)$ through a prompt of type $(x_1, f(x_1), ..., x_i)$. We refer to that prediction as $\hat{f}(x_i)$. To train the model $\mathcal{L}$ to ICL, we looked for a $\theta^*$ that optimizes the following auto-regressive objective:

$$\theta^* = \arg\min_{\theta} \mathbb{E}_{x_i \in D_I, f \in D_F} \left[ \sum_{i=0}^{k} l\left( f\left(x_{i+1}\right), \mathcal{L}_{\theta}\left((x_1, f(x_1), ..., f(x_i), x_{i+1})\right)\right) \right]$$

where $\mathcal{L}_{\theta}$ is a learner, $l : (y, \hat{y}) \to ||y - \hat{y}||^2$ is squared error and $f : x \to ax + b$ is a linear function with $a, b$ chosen at random according to some training distribution for functions $D_F$ and samples $x_i$ picked randomly according to a training distribution for points $D_I$. To simplify, we will note that $f \in D_F, x \in D_I$. We choose at random a function $f \in D_F$ and then a sequence of points $x_i \in D_I$ at random, random prompts, from a distribution $D_I$ at each training step. We update the model through a gradient update. We use a batch size of 64 and train for 500k steps. The models saw over 1.3 billion training examples for each distribution we studied. For $D_F$ and $D_I$ we used several distributions: the normal distribution $N(0,1)$, "rectangle" or uniform distributions over given intervals and bimodal distributions.

In comparing how model performance evolves with parameters like the number of layers of the model or number of attention heads, we tested the models on a variety of test distributions for both functions $D_F^t$ and data points or prompts $D_I^t$. But while in train we always take the same distribution $(D_F = D_I)$, in test, we sometimes take $D_F^t \neq D_I^t$. To see how the model performs in ICL relative to $(D_I^t, D_F^t)$, we generate a set of $N = 100$ functions in $D_F^t$; and our data samples for test are composed of $N_b = 64$ batches, each containing $N_p = 41$ points in $D_I^t$. In each batch b, for all points, we predict for each $x_k^b$, $k \geq 2$, $f(x_k^b)$ given the prompt $(x_1^b, f(x_1^b), ..., x_{k-1}^b, f(x_{k-1}^b), x_k^b)$.

---

[1]Our code follows that of Garg et al. (2022) and can be found in https://anonymous.4open.science/r/incontext-learning-556D/

We calculate for each function the mean average over all the points $N_p$ of all batches $N_b$, then do a mean average over all functions. Formally this is:

$$\epsilon_\sigma = \frac{1}{N}\Sigma_{i=1}^N \Sigma_{b=1}^{N_b} \frac{1}{N_b}\left(\frac{1}{N_p}\Sigma_{i=3}^{N_p}(pred_i^b - y_i^b)^2\right)$$

We define *error rate* $r_\epsilon = \frac{\epsilon_\sigma}{|\epsilon_* - \epsilon_0|}$ where $\epsilon_*$ is the best $\epsilon_\sigma$ error for a model M with $\hat{f}(x)$ calculated with Least Squares, and $\epsilon_0$ is the worst $\epsilon_\sigma$ error for a model $M$ such that $\hat{f}_M(x) = 0$, $\forall x$. In all our error calculations, we exclude the first two predictions of each batch from the squared error calculation, since we need at least two points to be able to find a linear function and the first two predictions by the model are hence almost always wrong.

### 4.1 MODELS DO NOT IMPLEMENT LINEAR REGRESSION

When trained on $D_F = D_I = N(0,1)$ and the target functions had values in [-1, 1], even small models were able to converge to a 0 average error. The error was not always identical to 0 at least in some batches but rather similar to Liu et al.'s finding on MSE estimation by transformers.

On the other hand, all the models had systematic and non 0 average error once we chose the target $f \in D_F^t = N(0,\sigma)$ for $\sigma > 2$. Figure 1 shows that the error rate increases substantially and non-linearly as $D_F^t = N(0,\sigma)$ and $\sigma$ increases. To ensure that comparisons between models are meaningful, for each $N(0,\sigma)$, we set a seed when generating the 100 random linear functions, ensuring that each model sees the same randomly chosen functions and the same set of prompting points $x_i$. The table 2 in the Appendix contains the full figures for average error.

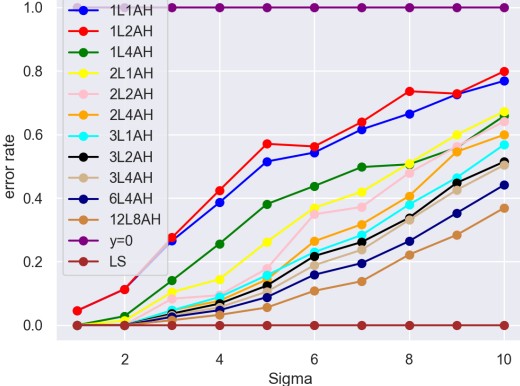

Figure 1: Evolution of error rates for various models with $D_F, D_I = D_I^t = N(0,1)$ and $D_F^t$ for various $N(0,\sigma)$. The black curve illustrates a model that predicts $f(x_n) = 0, \forall f$ and $\forall x_n$. The cyan line LS represents linear or ridge regression, which is trivially a perfect estimator given our totally clean input data.

The results in Figure 1 and Table 2 confirm that at least the larger models are able to generalize somewhat to unseen examples, given that all the curves in Figure 1 have lower error rates than the baseline that predicts $\hat{f}(x_n) = 0$ everywhere. But their generalizing ability was far from perfect; and contrary to what Akyürek et al. (2022); Von Oswald et al. (2023) have suggested, the models did not use linear regression to ICL the target function. If they had, we would not see the error patterns we do.

Our results are also quite different from Zhang et al. (2024), who say shifting the distribution sampled at inference of the functions does not affect their models. Our results show such a shift affects the results in an important way, where we take $N(0,1) = D_F$ ( but $D_F^t = N(0,\sigma)$ for $1 \leq \sigma \leq 10$. Figure 1 clearly shows that for transformer models with soft attention, this task shift reduces performance dramatically.

Giannou et al. (2024) also only examine differences in sampling the sequences of points in the prompt; i.e. in our notation $D_I \neq D_I^t$. We comment on this in Section 4.3.

## 4.2 REPLICATING SECTION 4.1 RESULTS FOR MODELS TRAINED ON OTHER DISTRIBUTIONS

We've just examined the behavior of models on test sampling from $N(0, \sigma)$ for larger $\sigma$ when the distribution of training data follows a simple Gaussian $N(0, 1)$. Our models, for any number of layers and attention head, have the same behavior when trained on different distributions but tested on $N(0, \sigma)$; they give good results when $D_F^t = D_I^t = N(0, 1)$, but offer degraded performance when tested on $N(0, \sigma)$ for larger $\sigma$.

**Training on bimodal distributions** We tested how our models fared with the bimodal distribution of training data, $0.5N(-1, 1) + 0.5N(1, 1)$. This increased the values of $f(x)$ the model can see during training.

Most of the models we tested had more robust performance with a bimodal distribution for $D_F = 0.5N(-1, 1) + 0.5N(1, 1)$ than they did with $D_F = N(0, 1)$ at least with $D_F^t = D_I^t = N(0, \sigma)$ and $n \geq 6$. The best models had almost equally good performance on $D_F^t = N(0, \sigma)$ for $\sigma \leq 3$ and superior performance with $D_F^t = N(0, \sigma)$ for $\sigma \geq 3$, as can be seen from Table 1. For the values of the table, we took $D_I^t = N(0, 1)$. The fact that performance varies with the distribution should not happen, if the models were using gradient descent to compute linear regression in ICL.

**Training on uniform distributions** We next trained our models on uniform distributions, in particular $U(-5, 5)$. This gives more control on the notion of maximum and minimum values the models see in training. Given the observations of Section 4.1 concerning the errors our models made on functions with large coefficients, we wanted to study whether these errors arose because the models hadn't encountered functions with such large coefficients in pretraining. By keeping $D_F, D_I$ normal or bimodal, we can't control "the largest value the model could see", because it's always possible that it could have generated a large value during training. By training on a uniform distribution, however, we know exactly what the smallest and largest values that the model could have seen in its training. For example, setting $D_F, D_I$ to $U(-5, 5)$, the largest value the model could have seen is $30 = 5 * 5 + 5$ and the smallest value it could have seen is $-30$. Most likely it saw values significantly $> -30$ and $< 30$.

Training with $U(-5, 5)$ gave good results for $D_F^t = D_I^t = U(-1, 1)$. Models were able to find target functions with coefficients in [-1,1] from only 2 points (see leftmost plot of Figure 9 in Appendix C); and all our models work well when $D_F, D_I, D_F^t, D_I^t$ use the same distribution. The models trained on a uniform distribution sometimes do even better than models trained on N(0,1) or a bimodal distribution–up to three times better for $D_F^t = D_I^t = N(0, 9)$ as Table 1 shows. Learning was at times very efficient, requiring just two prompts, as in Figure 9 (Appendix B).

## 4.3 ERROR ANALYSIS, SIGMOID APPROXIMATIONS AND BOUNDARY VALUES

Our models' performance depends on how often it has seen examples "similar" to the target function value it is trying to predict. At first, we thought this was due to the choice of coefficients in the target function $f(x) = ax + b$. However, experimentally, we verified that this is really due just to the values in the sequences it has seen. Extreme examples for $D_F = N(0, 1)$ with tests in $[100, 101]$ are in figure 2. In Appendix C we illustrate quantitatively intervals $I$ within which models have seen a large majority of values of sequences given a different training regime. Given a pretraining with over a billion examples, models will have seen prompts for functions with outside of $I$, just not many of them. As the models are tested with $D_F^t = N(0, \sigma)$ and so required to predict $\hat{f}(x)$ for $f(x) \notin [-2, 2]$, all the models do less and less well; Figure 5 in the Appendix shows similar behavior for models trained on uniform distributions.

This motivated us to investigate errors our models made for target functions $f(x) \notin [-2, 2]$–i.e. the values of $\hat{f}(x)$ outside the interval that includes the vast majority they have seen. Our models exhibit problematic behavior of 2 kinds. Even our best models, for $f(x) \notin [-2, 2]$ but reasonably close, say in $[-9, 9]$, predict $\hat{f}(x)$ to a sigmoid-like function with correct estimates for the target function within a certain interval. Consider the middle plot for $f(x) = 10x$ in Figure 2. The plot shows

| models / $\sigma$ | 1 | 2 | 3 | 4 | 5 | 6 | 7 | 8 | 9 | 10 |
|---|---|---|---|---|---|---|---|---|---|---|
| $3L4AH_N, d_{emb} = 64$ | 0.0 | 0.0 | 0.22 | 0.4 | 1.73 | 6.56 | 8.56 | 20.44 | 39.73 | 53.93 |
| $3L4AH_B, d_{emb} = 64$ | 0.03 | 0.15 | 0.53 | 1.32 | 2.74 | 3.91 | 5.52 | 10.22 | 13.86 | 22.72 |
| $3L4AH_U, d_{emb} = 64$ | 0.02 | 0.03 | 0.13 | 0.36 | 0.84 | 1.79 | 2.54 | 7.06 | 11.38 | 17.75 |
| $6L4AH_N, d_{emb} = 64$ | 0.0 | 0.0 | 0.2 | 0.38 | 1.58 | 5.72 | 7.99 | 15.53 | 32.96 | 50.35 |
| $6L4AH_B, d_{emb} = 64$ | 0.01 | 0.04 | 0.23 | 0.44 | 1.19 | 2.15 | 3.08 | 4.8 | 9.98 | 18.01 |
| $6L4AH_U, d_{emb} = 64$ | 0.02 | 0.04 | 0.11 | 0.24 | 0.57 | 1.36 | 1.82 | 4.62 | 10.23 | 15.07 |
| $12L8AH_N, d_{emb} = 256$ | 0.0 | 0.0 | 0.32 | 1.34 | 3.14 | 8.8 | 12.13 | 30.14 | 49.37 | 73.93 |
| **sorted** $12L8AH_N$ | 0.0 | 0.01 | 0.32 | 1.63 | 3.69 | 8.39 | 10.06 | 27.11 | 43.23 | 58.56 |
| $12L8AH_B, d_{emb} = 256$ | 0.0 | 0.01 | 0.08 | 0.29 | 0.78 | 2.23 | 3.66 | 9.04 | 18.68 | 30.23 |
| **sorted** $12L8AH_B$ | 0.01 | 0.03 | 0.18 | 0.25 | 0.74 | 2.27 | 2.62 | 6.87 | 13.73 | 20.8 |
| $12L8AH_U, d_{emb} = 256$ | 0.0 | 0.01 | 0.13 | 0.71 | 1.92 | 6.78 | 10.92 | 27.91 | 38.75 | 64.39 |
| **sorted** $12L8AH_U$ | 0.01 | 0.01 | 0.13 | 0.75 | 2.12 | 6.18 | 10.5 | 26.8 | 36.3 | 53.48 |
| $REF_{D_F^t, D_I^t}$**: y=0** | 1.52 | 4.43 | 13.55 | 19.94 | 30.81 | 44.75 | 52.71 | 76.11 | 105.43 | 128.52 |

Table 1: Comparison showing the evolution of squared errors for models trained on different distributions; index N: $D_F = N(0,1)$, B $D_F = 0.5N(-1,1) + 0.5N(1,1)$ and $D_F = U(-5,5)$. We show error rates for models prompted without and with the natural ordering on the prompts [sorted], for the large model size. $D_i^t = U(-1,1)$ and $D_F^t = N(0,\sigma)$

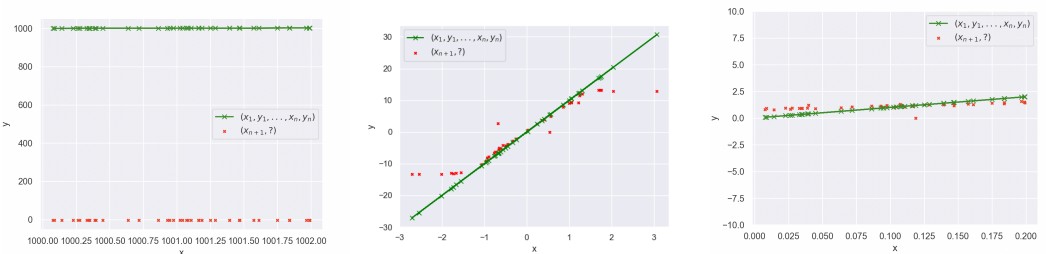

Figure 2: Plots for model 12L8AH, trained on $D_I = D_F = N(0,1)$ for $f(x) = x$ for high values (left) of $x$ and $f(x) = 10x$ for normal (middle) then for low values of $x$ (right)

that the model's prediction $\hat{f}(x)$ diverges dramatically from $f(x)$ outside of a certain interval, but the rightmost plot shows that it has approximated well within that interval. Appendix D contains a graph over length of the prompt showing that it has learned something with ICL.

For equations $f(x)$ sampled outside $N(0,1)$ (for example $f(x) = 30x + 30$ and $D_I^t = N(0,1)$, however, the results are catastrophic and similar to those in the first plot of Figure 2. Figure 4 in the Appendix shows that the model doesn't converge to any stable prediction with ICL.

This behavior across a wide range of models. For example with $D_F = D_I = U(-5,5)$, consider again as an illustrative example the target function, $f(x) = 9x$ for our largest trained model. The model approximates $f(x)$ well within a certain range $[-B, B]$, but it predicts $\hat{f}(x)$ to be a constant function for $x$ such that $\hat{f}(x) \notin [-B, B]$ within a certain range (See Figure 5 and discussion in Appendix C). We call values $-B, B$ *boundary values*. By training on uniform distributions, we can determine the boundary values exactly; e.g, for $U(-5,5)$ $B = 5 \times 5 + 5$. These are the biggest and smallest values the model could have seen during training. If such a model hasn't seen a value above B or below -B, it won't infer one. Different models trained on different uniform distributions give different boundary values (see below).

All our models trained on $U(-5,5)$ estimate the target function more or less well for $x$ with $f(x) \in [-30, 30]$ ; but once we are outside $[-B, B]$, the estimations become constant functions or chaotic. Figure 5 with equation $f(x) = 40x + 40$—illustrates this chaotic behavior as does the leftmost plot of Figure 2 for function $f(x) = x$ with large number inputs.

To summarize, we observed the following: **Empirical Generalization** For all models $M$ and for values $B < f(v) < B + \alpha$, where $\alpha$ is a constant determined by $M$, $fh_M(v) \approx B$, and for $-B - \alpha < f(v) < -B$, $fh_M(v) \approx -B$. However for functions and data samples when the values of $f(x)$ in the prompt sequence are such that $f(x) > B + \alpha$ or $< -B - \alpha$, the model assigns $\hat{f}(v)$ random values for $\hat{f}(v)$ far away from $B$ (i.e $> B + \alpha$ or $< -B - \alpha$.

Constraints from boundary values hold for all transformer models tested (for plots see Appendix D and Figure 6) and for attention only models (See Appendix D, Figure 8). However, due to the parameter $\alpha$, larger models trained on the same distribution and the same number of data will ICL $\mathcal{L}$ functions over a slightly larger number of intermediate values than smaller models, as Figure 1 suggests. Figure 7 in the appendix shows plots for the predictions of two models (12L8AH, and 6L4AH) for $D_F, D_I = N(0, 1)$ for target $f(x) = 10x$. The larger model has boundary values $\approx$ -13.7, 13.7, the smaller one boundary values $\approx$ -12, 12.

Giannou et al. (2024) also noted something like boundary values with their linear transformer architecture but they do not accord them much importance. They also investigated out of distribution behavior but only on $D_I \neq D_I^t$ (covariate shifts in Zhang et al. (2024)) (not shifts from $D_F$). They found that after 4 layers transformer model performance did not perform. We found that larger models did improve performance, but when we set $D_I \neq D_I^t$, we got bad results when the function's values on those points were outside what we call boundary values, something which held for all models.

Zhang et al. (2024)'s covariate shift is also different from our experiments. They shift the prompt distribution but not that of the query. When we take a distribution over input points in train $D_I$ and set $D_I^t \neq D_I$, our shift is not the same; we shift both prompt and query distributions. With covariate shifts we found that the choice of points is important and model performance degrades considerably when the values of the functions on the chosen points lie beyond what we call boundary values, which Zhang et al. (2024) do not. As far as we know we are the first to take boundary values and their dependence on model parameters as key indications of what is actually going on in ICL.

.

### 4.4 Predictions for models with only attention layers or with only MLP

To understand better which components in the transformer architecture are responsible for ICL, we tested various components. We found that attention layers (AL) were the important components for ICL but ICL only worked reasonably well when the model had 2 AL (see also figure 4). Beyond 2 AL what mattered most was the number of attention heads (whether they are summed over all layers or counted within a layer). A single AL model had only a very limited ICL generalization capability beyond testing on $D_F^t = N(0, 1)$, but it did better than a 12 layer MLP, which showed no ICL capability. Attention-only models could ICL linear functions reasonably well, at least in when $D_F = D_F^t$; the large 2 attention only layer model with 32 AH was more robust than the full transformer model with 1 (AL and MLP layer) and 1 or 2 AH (See Table 2 Appendix B). Tables 3 4 in Appendix and Figure 3 give details of various AL models on normal and uniform distributions.

### 4.5 Ordering prompts and restricting their size

Model performance improves when the sequence of prompts for the $x_i$ are sorted to follow the natural order on $\mathbb{R}$, especially for bigger models. Error rates compare to error rates without sorting for small values of $\sigma$ with $D_F^t = N(0, \sigma)$ and are lower by up to a third on other test values, depending on the training distribution (see Table 1).

While at least 2 points are needed to find a linear function, all model performance regardless of training distribution degrades when the size of the prompt during inference is greater than the maximal size of prompts seen in training, as the rightmost plot in Figure 9 shows (Appendix E). Further models did better with the distributions that were exactly the size (41 data points) of those in their training We tested a 12L8AH model with with smaller sequences in a kind of "curriculum learning" and without curriculum; we found that the model without curriculum training performed better. All this implies that a model takes into account the whole sequence in its calculations, not just the last two or three data points. Had the model only looked at a small fixed subsequence, larger sized

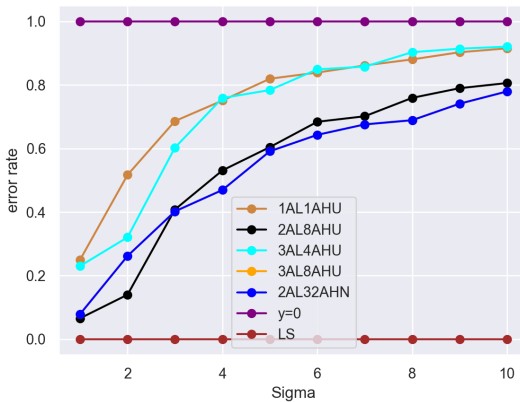

Figure 3: Evolution of error rates models with attention layers only. We give figures for a model with only 1 attention layer/1AH (1AL1AH) two 2-attention layer only models (2AL8AH, 2AL32AH) and two 3 attention layer only model (3AL4AH,3AL8AH). $D_I = D_F = U(-1,1)$, $D_i^t = U(-1,1)$ and $D_F^t = N(0,\sigma)$. All models have embeddings of size 64, except $2AL32AH$ has size 256.

prompts in inference would not have affected model performance and curriculum learning should have improved it.

## 5   WHAT AND HOW ARE THE MODELS LEARNING?

The hypotheses and theoretical constructions of Akyürek et al. (2022); Von Oswald et al. (2023) led us to expect that a transformer model given $(x_1, f(x_1), ..., x_n, ?)$ would perform a linear regression to ICL a linear function. In this case, the models should generalize without difficulty. But this is not what we observed. Error rates depend on the distance of the target function's values from the majority of the data points in the model's training. Models are also sensitive to the entire sequence of ICL prompts, not just the minimal number needed to compute a linear function. Error analysis showed the existence of boundary values $-B, B$; models do well on the interval $[-B, B]$ degrade outside of them. These boundary values fluctuate depending on model training distributions and size. All this is strong evidence that models did not learn to use linear regression to solve this task and failed to learn the concept of a strictly monotone increasing or decreasing linear function in $\mathcal{L}$ over arbitrarily large or at least many large intervals of $\mathbb{R}$.[2]

The lack of generalizability might suggest our models overfit the data. However, the pretraining data has no noise, and it's too large to be memorized by our models (our largest models with 256 size embeddings have $< 10^7$ parameters; each parameter would have to encode on average sequences for over 100 different functions). Moreover, our models performed similarly on several different training distributions for $D_F$ and $D_I$ and tested on $N(0,\sigma)$ for $\sigma \in \{1, 2\}$. Given that 100 samplings with $D_F^t = N(0,1)$ nets on average 20 functions with coefficients the model with $D_F = D_I = U(-1,1)$ has not seen in training, we would expect the model's performance to degrade more substantially than it did. This implies that the models didn't overfit to their training regimes.

Rather than computing a linear function in this task, the models estimate continuations of sequences based on sequences they have seen. This is in line with Olsson et al. (2022)'s finding that a copying and comparison mechanism (induction head) is at the heart of ICL. They show that induction heads only exist for attention-only models with two or more layers and that larger models' induction heads can exploit sequences that are "more dissimilar" to each other than smaller models can.

Our *induction head hypothesis* is that a model predicts a value for $f(x_n)$ given a prompt sequence $\vec{x} = (x_{1,1}, x_{1,2}(= f(x_1)), x_{2,1}, x_{2,2}, ...x_{n,1}, ?)$ by using a projection from similar sequences or

---

[2]This makes sense in terms of Asher et al. (2023)'s characterization of learnability. The concept of a strictly monotone increasing or decreasing linear function describes a $\Pi_1^0$ set in the Borel hierarchy which Asher et al. (2023) show is not learnable using ordinary LLM assumptions.

subsequences in the training, $\vec{y} = (y_{1,1}, y_{2,2}...y_{n,1}, y_{n,2})$, with $x_{i,1}$ close to $y_{i,1}$ for some $j$ and $x_{i,2}$ close to $y_{j,2}$. The effects of prompt length on performance imply that the whole sequence matters with $p_2 \leq p_1$ for optimal predictions. he fact that the larger models with more attention heads respond well to well-ordered prompts suggests that they can exploit comparing sequences that converge or diverge from the target sequence $\vec{x}$ in different ways as the prompts $x_{i,1}$ near $x_{n,1}$ increase or decrease. This is evidence for the pointwise comparison we are proposing (which is more complicated and potentially more accurate than simply averaging the $y_{n,2}$ of the three closest $y_{n,1}$ neighbors of $x_{n,1}$) (cf. Olsson et al. (2022)).

Our observations about boundary values provide further empirical support for a particular induction head hypothesis. Given boundary values, $-B, B$, all or the vast majority of the sequences the model has seen have values $z_i$ with $-B < z_i < B$. If the target sequence $\vec{x}$ has maximum values $-B < x_i < B$, i.e. $-B < Maxval_{x_i}\vec{x} < B$, then chances are high that the model will find a weighted set of sequences $Y$ close to the test sequence $\vec{x}$ and compute bounds for $x_{n,2} = f(x_n))$.

We now offer a mathematical model of the projection. We assume the standard measure over sequences. Let $\vec{x}$ be the sequence generated by the target linear function $f$. To icl $f$, a model must construct a function $\mathfrak{h}(Y_{\vec{x}}, \vec{x})$ that computes a distance $d$ between the values it has seen in $Y_{\vec{x}}$ and the targets $\vec{x}$ for some optimized set $Y_{\vec{x}}$ of sequences close to $\vec{x}$. If $\mathfrak{h}(Y_{\vec{x}}, \vec{x})(x_{k,1}) = z_{k,2}$ is the k-th member of $\mathfrak{h}(Y_{\vec{x}}, \vec{x})$, we optimize $\mathfrak{h}$ such that $|z_{k,2} - x_{k,2}|$ is minimized for all $k$. The model then averages these distances to yield an "average" $\mathfrak{h}(Y_{\vec{x}}, \vec{x})$ to compute $z_{2,n} = \hat{f}(x_{1,n})$.

In sum, a model $M$ computes $\hat{f}_M$ via:

$$\hat{f}(x_n) = x_{n,2} = \frac{1}{n}\sum_{i=1}^{n}\mathfrak{h}(Y_{\vec{x},x_i})(x_{n,1}), \, for \, -B < Maxval_{x_i}\vec{x} < B$$

$$and \, \hat{f}(x_n) \approx B(-B), \text{if } Maxval_{x_i}\vec{x} < -B - \alpha_{\mathcal{L}}, \, or \, Maxval_{x_i}\vec{x} > B + \alpha_{\mathcal{L}}$$

$$Otherwise \, \hat{f}(x_n) \, takes \, a \, random \, value \in [-B, B], \, \alpha_{\mathcal{L}} > 0 \, a \, characterstic \, model \, value$$

According to our projection, the larger the set of close $\vec{y} \in Y_{\vec{x}}$, the better the projection and the prediction. For prompts outside the boundary values $-B, B$, the closest $\vec{y}$ are those with values near the boundary ($y_{n,2} \approx B(-B)$). Using our projection, the model $M$ will predict $x_{n,2} \approx B(-B)$; once $x_{n,1}$ is very far away from known data points, the averaging method will just give some value in $[-B, B]$. It also predicts that model performance will be sensitive to a choice of training distribution for $D_F, D_I$ as well as a choice of test distributions. Our projection also explains why training a model without curriculum does better than a model with curriculum: it can see more relevant steps.

Our formulation of the projection thus accords with our empirical observations, and the weighted averages are calculable in a 2 layer Attention only model with suitable heads. The induction head hypothesis is less precise then linear regression but can approximate it given an appropriate set $Y$.

## 6 CONCLUSION

In this paper we have shown a systematic failure case of decoder-only transformer models of various sizes (up to 9.5 million parameters) and architectures. All models failed to learn robustly the class of linear functions on non-noisy data, a task which is entirely determined by only two points and involves a trivial mathematical operation shown by construction to be learnable by LLMs. However, the models did learn something different that enabled them to approximate linear functions over intervals where their training gave lots of examples. Rather than learning a standard algorithm for the task, these models instead perform a projection from close sequences seen during training.

Our investigations perforce focus on relatively small models, but they highlight a broad issue with ICL: the gap between what LLMs *can* learn and what they *actually* learn. Larger models also face this limitation. The minimality of our examples and the capacity to easily train the models from scratch is a key strength of our study.

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

## A  TRAINING DETAILS

**Additional training information:** Like Garg et al. (2022), we use also the Adam optimizer Diederik (2014) , and a learning rate of $10^{-4}$ for all models.
**Computational resources:** We used 1 GPU Nvidia Volta (V100 - 7,8 Tflops DP) for every training involved in these experiments.

## B  ERROR PROGRESSION FOR MODELS TRAINED ON $N(0,1)$ DISTRIBUTIONS TESTED ON $N(0,\sigma)$

When $D_I = D_F = N(0,\sigma)$ there is for $x \in N(0,\sigma)$ an over 85% chance of $f(x) \in [-4\sigma^2 - 2\sigma, 4\sigma^2 + 2\sigma]$ and a 95% chance $f(x) \in [-2\sigma, 2\sigma]$. So a model with $\sigma = 1$ $D_F = D_I = N(0,1)$ has seen sequences of values for $f$ with $f(x) \in [-2, 2]$ more than 95% of the time.

| models / $\sigma$ | 1 | 2 | 3 | 4 | 5 | 6 | 7 | 8 | 9 | 10 |
|---|---|---|---|---|---|---|---|---|---|---|
| 1L1AH $d_{embedding}$=64 | 0.1 | 0.8 | 5.1 | 13.1 | 26.9 | 39.7 | 53.0 | 84.8 | 120.0 | 153.2 |
| 1L2AH $d_{embedding}$=64 | 0.1 | 0.8 | 5.3 | 14.4 | 29.8 | 41.1 | 55.0 | 93.8 | 120.4 | 159.2 |
| 1L4AH $d_{embedding}$=64 | 0.0 | 0.2 | 2.7 | 8.7 | 19.9 | 32.0 | 42.8 | 64.5 | 92.3 | 131.2 |
| 2L1AH $d_{embedding}$=64 | 0.0 | 0.1 | 2.0 | 4.9 | 13.7 | 27.0 | 36.1 | 64.9 | 99.0 | 134.0 |
| 2L2AH $d_{embedding}$=64 | 0.0 | 0.0 | 1.6 | 3.2 | 9.3 | 25.5 | 32.0 | 61.1 | 92.9 | 127.8 |
| 2L4AH $d_{embedding}$=64 | 0.0 | 0.0 | 0.9 | 2.6 | 7.5 | 19.3 | 27.3 | 51.8 | 90.2 | 119.4 |
| 3L1AH $d_{embedding}$=64 | 0.0 | 0.0 | 0.9 | 3.0 | 8.2 | 16.8 | 24.4 | 48.4 | 76.7 | 113.2 |
| 3L2AH $d_{embedding}$=64 | 0.0 | 0.0 | 0.7 | 2.3 | 6.5 | 15.9 | 22.5 | 43.1 | 74.0 | 102.5 |
| 3L4AH $d_{embedding}$=64 | 0.0 | 0.0 | 0.6 | 1.9 | 5.5 | 13.8 | 20.4 | 42.2 | 70.3 | 100.4 |
| 6L4AH $d_{embedding}$=64 | 0.0 | 0.0 | 0.5 | 1.6 | 4.6 | 11.6 | 16.8 | 33.7 | 58.3 | 87.9 |
| 12L8AH $d_{embedding}$=256 | 0.0 | 0.0 | 0.3 | 1.1 | 2.9 | 7.9 | 11.9 | 28.3 | 46.9 | 73.5 |
| **REF: y=0** | 2.19 | 7.05 | 19.22 | 33.94 | 52.23 | 73.08 | 86.02 | 127.43 | 165.27 | 199.31 |

Table 2: Comparison to show the evolution of squared $\epsilon$ type error depending on the distribution according to which we take the parameters, without taking into account the error of the prediction of the first and second prompts. $D_i^t = N(0,1)$

## C  Plots for boundary values with $N(0,1)$ and $U(-5,5)$

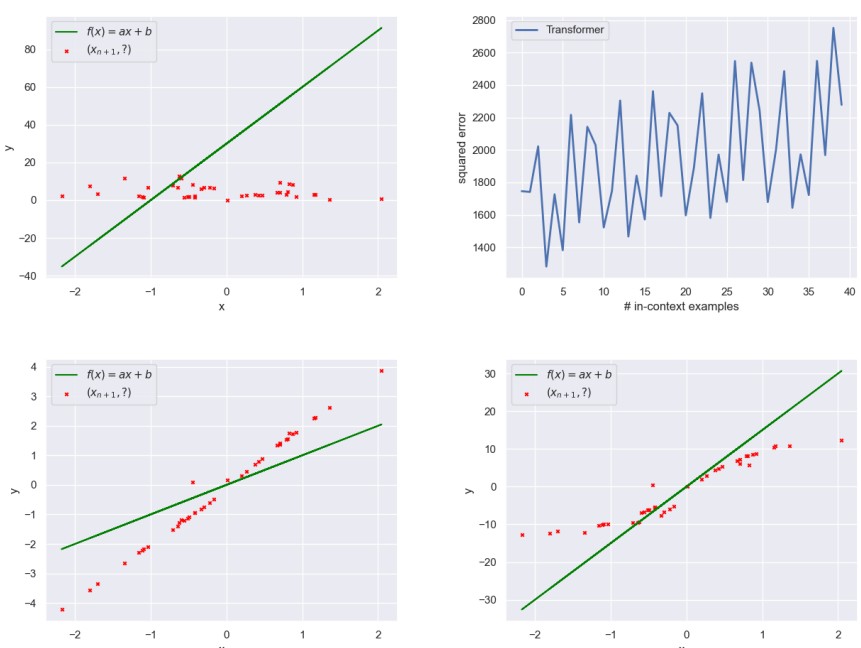

Figure 4:  Plots on first line of predictions for the 12L8AH model trained on $N(0,1)$ and error evolution over number of prompts for $f(x) = 30x + 30$. On second line Plots for $f(x) = x$ and $f(x) = 15x$ for models 2L attention only with 32AH and $d_{embedding} = 256$

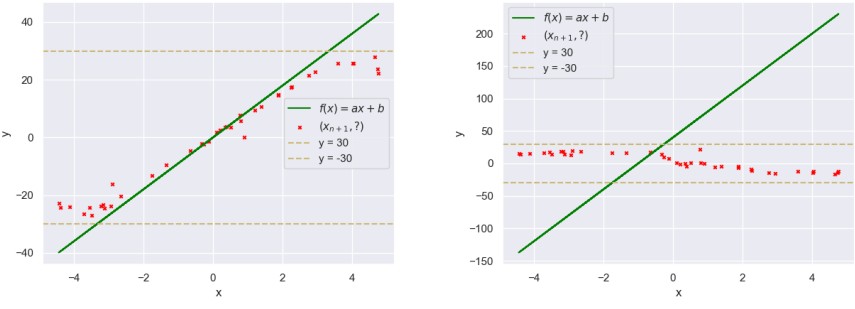

Figure 5: Plots for $f(x) = 9x$ and $f(x) = 40x + 40$ for a 12l8ah model trained on $U(-5,5)$

As shown in the left plot in Figure 5, $\hat{f}^+(v) \approx 30$ for values $v$ for which the ground truth target function $f$ is such that $30 \leq f(v)$, and the model predicts an approximally constant function $\hat{f}^-(v) \approx -30$ for values $v$ on which $f(v) \leq -30$. Given a training on $U(-5,5)$ we can calculate 30 and -30, with $30 = 5*5+5$ and $-30 = -5*5-5$, to be the boundary values for the models there.

## D  Example of boundary values for attention only models

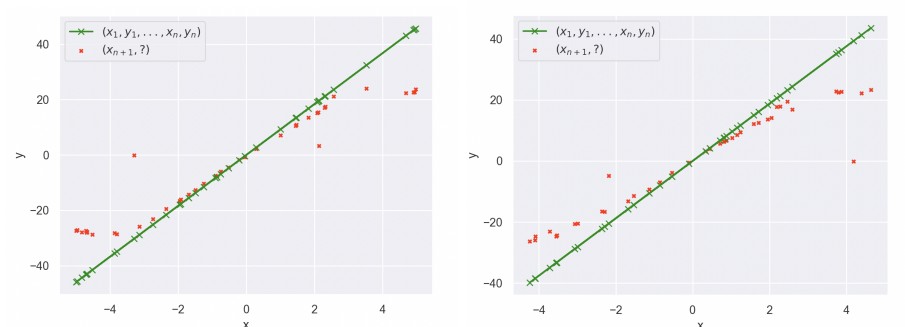

Figure 6: Boundary values: Plots for $f(x) = 9.4x$ for models 3L4AH and 6L4AH, $D_I = D_F = D_I^t = D_F^t = U(-5, 5)$

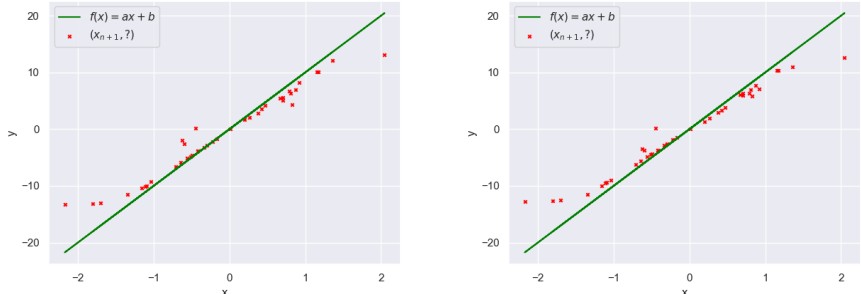

Figure 7: Plots for $f(x) = 10x$ by a 12L8ah model and by a 6L4ah model.

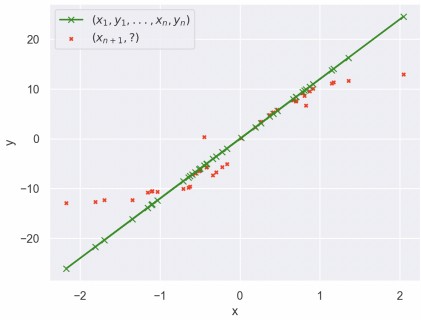

Figure 8: Boundary values for 2L32ah attention only model, with $d_{embedding} = 256$ to ICL the function $f(x) = 12x$

# E  FAILURE TO GENERALIZE TO LONGER PROMPT SEQUENCES: FIG9

| models / $\sigma$ | 1 | 2 | 3 | 4 | 5 | 6 | 7 | 8 | 9 | 10 |
|---|---|---|---|---|---|---|---|---|---|---|
| $1AL1AH_U$ | 0.38 | 2.29 | 9.3 | 14.97 | 25.25 | 37.54 | 45.4 | 67.0 | 95.19 | 117.6 |
| $2AL8AH_U$ | 0.1 | 0.62 | 5.53 | 10.59 | 18.62 | 30.61 | 36.97 | 57.79 | 83.26 | 103.58 |
| $3AL4AH_U$ | 0.35 | 1.42 | 8.17 | 15.13 | 24.15 | 37.99 | 45.2 | 68.73 | 96.37 | 118.3 |
| $3AL8AH_U$ | 0.12 | 1.16 | 5.45 | 9.36 | 18.22 | 28.77 | 35.62 | 52.44 | 78.12 | 100.18 |
| $2Al32AH_N$ | 0.06 | 0.91 | 5.96 | 10.43 | 18.96 | 30.11 | 36.77 | 55.59 | 81.66 | 103.17 |
| $REF_{D_F^t, D_I^t} : y = 0$ | 1.52 | 4.43 | 13.55 | 19.94 | 30.81 | 44.75 | 52.71 | 76.11 | 105.43 | 128.52 |

Table 3: Comparison showing the evolution of squared errors for models with attention layers only. We give figures for a model with only 1 attention layer/1AH (1AL1AH) two 2-attention layer only models (2AL8AH, 2AL32AH) and two 3 attention layer only model (3AL4AH,3AL8AH). $D_I = D_F = U(-1,1)$, $D_i^t = U(-1,1)$ and $D_F^t = N(0,\sigma)$. All models have embeddings of size 64, except $2Al32AH$ has size 256.

| models / $\sigma$ | 1 | 2 | 3 | 4 | 5 | 6 | 7 | 8 | 9 | 10 |
|---|---|---|---|---|---|---|---|---|---|---|
| $1L1AH_N$ $d_{embedding}$=64 | 48.8 | 57.62 | 73.48 | 84.51 | 116.63 | 129.52 | 142.34 | 177.69 | 191.05 | 246.43 |
| $2L8AH_N$ $d_{embedding}$=64 | 2.24 | 4.81 | 5.8 | 7.19 | 10.01 | 19.04 | 30.22 | 38.03 | 73.32 | 118.89 |
| $2L32AH_N$ $d_{embedding}$=256 | 1.17 | 2.64 | 3.47 | 5.01 | 7.88 | 16.85 | 24.1 | 40.98 | 66.04 | 95.03 |
| **REF: y=0** | 2.19 | 7.05 | 19.22 | 33.94 | 52.23 | 73.08 | 86.02 | 127.43 | 165.27 | 199.31 |

Table 4: Comparison to show the evolution of squared $\epsilon$ type error depending on the distribution according to which we take the parameters, without taking into account the error of the prediction of the first and second prompts. $D_F = D_I = D_i^t = N(0,1)$ for models with attention ONLY

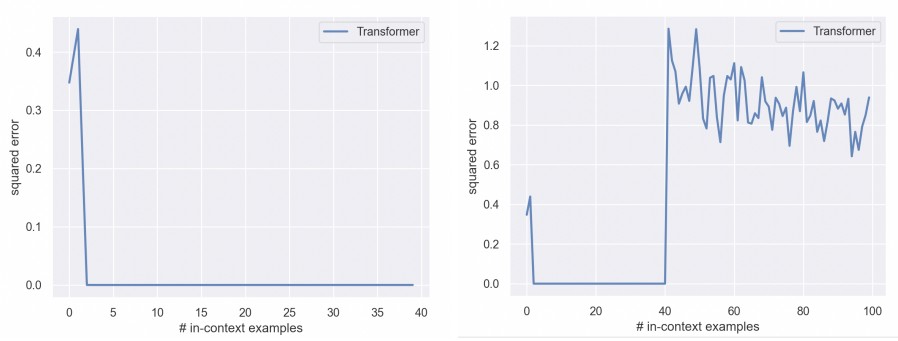

Figure 9: Plot of ICL for $f(x) = x$ with $D_F = D_I = D_I^t = U(-5,5)$ for the model 12L8AH; the one on the left is a zoom in on the first 40 points, where we see that models can often learn from 2 points, the second a view of what happens overall, when models are trained on sequences of length 41 prompts.

