# OpenReview forum: "Re-examining learning linear functions in context"
_ICLR.cc/2025/Conference — ICLR 2025 Conference Withdrawn Submission_

### Official Review · Reviewer_i5eC · 2024-10-31

**Soundness:** 2
**Presentation:** 3
**Contribution:** 2
**Rating:** 5
**Confidence:** 3

**Summary:**

The paper investigates in-context learning (ICL) across various training and testing scenarios using different sizes of transformer models trained from scratch. Building on previous work, it highlights systematic failures in these models' ability to generalize to data outside the training distribution, revealing some limitations of ICL.

**Strengths:**

1. The paper focuses on an important and challenging problem: understanding the in-context ability of language models.
2. The writing is clear and easy to understand.
3. The authors provide code and detailed instructions for reproduction.

**Weaknesses:**

1. The models and empirical studies in the paper differ significantly from current large language models, potentially creating a gap between the claims and reality.
2. The findings of the paper have been previously proposed in other works.
3. The paper is missing some key references.

**Questions:**

Can you explain how your findings differ from the following paper? In particular, [1] also discusses how distribution influences the in-context learning ability to learn linear functions.

[1] Trained Transformers Learn Linear Models In-Context.
[2] Transformers as Statisticians: Provable In-Context Learning with In-Context Algorithm Selection

---

> ### Author Response · Authors · 2024-11-13
>
> We thank the reviewer for the comments.
>
> We address the following three concerns
> 1- The models and empirical studies in the paper differ significantly from current large language models, potentially creating a gap between the claims and reality.
> 2- The findings of the paper have been previously proposed in other works.
> 3- The paper is missing some key references.
>
> As we mentioned we need small models to do training from scratch.  We need to train from scratch to avoid “leakage” from uncontrolled pretraining.  We did query larger models but we cannot train them from scratch.
>
> Thank you for pointing out these two references.  They are very helpful and we will certainly cite and discuss them in the final version.   Nevertheless with respect to [1], though similar to our work, [1] works with linear attention, whereas we look at attention layers as they actually are used with softmax (thus one could criticize [1] in the way you do in point 1 more than us at least on this score). In addition, [1] uses a new kind of optimization or training with gradients and a special fixed initial point.  This means that their architecture and training are quite different from what normally happens with transformers; they are interested in getting a revised transformer-like model to learn linear functions, while we want to find out whether transformers as they actually are learn linear functions or something else.   The results for the two architectures are quite different:  While [1] says task shift does not affect their models, our task shifts affect the results in an important way, where we take D^{train}_H = N(0,1) (Our D_F) but  D^{test}_H = N(0, \sigma) (our D^test_H) for 1 \leq \sigma \leq 10.  Figure 1 clearly shows that for transformer models with soft attention, this task shift reduces performance dramatically.   We also note unlike [1] that prompts that are too long induce chaotic behavior.
>
> In the covariate shift [1] also does something different from what we do.  In covariate shift in [1],  the distribution in the prompt is shifted but the distribution of the query stays the same.  We do something different.  When we take a distribution over input points in train D_I and set D^test_I \neq D_I, our shift is not the same; we shift both prompt and query distributions.  With covariate shifts we found that the choice of points is important and model performance degrades considerably when the values of the functions on the chosen points lie beyond what we call boundary values.  As far as we know we are the first to point out these boundary values and their dependence on model parameters.  At least [1] does not do this.  Our mathematical formulation of what models do, explains their behavior.  We will highlight this in the revised version.
>
> With respect to suggested reference [2], they largely follow what many other ICL papers do—offer a proof by construction that under certain assumptions transformers can implement many different algorithms for computing linear functions and other tasks as well.  Their empirical experiments show that under suitable training and testing distributions for sampling, transformer models can learn such algorithms. They signal something like our boundary values B, -B  and propose to ignore values outside [B, -B] by “clipping” them.  We take a very different approach.  By ignoring these outside values, we don’t really know what algorithm transformer models have implemented; we demonstrate with out of distribution robustness tests that transformer models don’t use any of the standard algorithms (ridge regression, linear regression,...) but do a different kind of projection and interpolation.
> We have included a short discussion of these 2 papers in the revised submission; hopefully the revised version will clarify and address your concerns.

---

> > ### Comment · Reviewer_i5eC · 2024-11-26
> >
> > Thank you for your response. Having considered your rebuttal and the other reviews, I maintain my original score.

---

### Official Review · Reviewer_2Uvx · 2024-11-01

**Soundness:** 2
**Presentation:** 1
**Contribution:** 2
**Rating:** 3
**Confidence:** 4

**Summary:**

The study investigates in-context learning (ICL) in transformer models, focusing on their ability to learn and generalize linear functions from contextual prompts. Inspired by previous work, the authors examine various transformer models, including small ones trained from scratch, to explore whether they can learn linear functions and generalize beyond the training distribution.

However, there are two main problems in this paper:

### 1. The writing problem: There are many typos, e.g., in ``line 047'', there should be a ''.'' after ''training data''.
### 2. Novelty: The paper indeed provides robust experiments to show the main point, but it lacks novelty, such as how to improve this problem.

**Strengths:**

The paper has the following strengths:

### 1. Clear Motivation: The paper begins with a well-defined motivation, addressing gaps in the current understanding of in-context learning (ICL) in transformer models, especially for generalization.

### 2. Comprehensive Experiments: The experiments cover various transformer architectures and test them on various distributions.

**Weaknesses:**

The paper has the following weaknesses:

### 1. Clarify Terminology and Notation: The writing is a little poor. For example, in ``line 047'', there should be a ''.'' after ''training data''. Furthermore, the table should be in a more beautiful structure.

### 2. Explanation for the Problem: Although the paper provides various experiments, it should explain the failures of these models to generalize to data not in the training distribution.

### 3. Novelty: The paper provides robust experiments to show the main point but lacks novelty, such as how to improve this problem.

**Questions:**

1.. Explanation for the Problem: Could you please explain the failures of these models to generalize to data not in the training distribution?
2. Could you please provide some methods to improve this problem?

---

> ### Author Response · Authors · 2024-11-13
> **officieal comment**
>
> We thank the reviewer for their comments.
>
> We will clarify our notation in the final version.
>
> With regard to an explanation of the failures of these models to generalize to data not in the training distribution, the last section explains that the model is doing something different and the mathematical model in the last section explains the model behavior.
>
> To answer your question concerning failure to generalize, please see the last section of the paper.  We have clarified our mathematical model and our notation in a revised version of our paper that we have put on the site.
>
> To improve the method is an important concern.  Of course you could program it to do linear regression.  The problem is that this isn’t learning the function class.  But we feel that learning linear regression is not really the issue here.  The issue is figuring out what transformers are doing in ICL for simple tasks so that we have a hope of understanding what they really do in more complex tasks.

---

> > ### Comment · Reviewer_2Uvx · 2024-11-24
> >
> > Thanks for the response. The response answers my questions. However, I still think the paper lacks novelty and is not ready to be published. Thus, I keep my score as 3.

---

> > > ### Author Response · Authors · 2024-11-24
> > > **reply to reply**
> > >
> > > Thanks for taking the time to read our response.  We very much appreciate that.
> > >
> > > Just to be clear about what we think our novel contribution is:  we show that the theoretical reconstructions that prevail  in the literature are not what transformers are doing when they icl linear functions.  Models do not compute values of a linear function via linear regression, ridge regression, etc; if they did the performance would not show degradation on sequences that are rare.  We also observed two kinds of degradation that depend on parameters we call boundary values.  As far as we know, we are the first to talk about such values.  Finally, the models use the whole sequence and only sequences of a certain length to do the task.  We conclude from these observations that the models do not understand the sequence as the plot of a function with parameters to be estimated but rather interpolate values from other similar sequences.  The method the models discover is ingenious and works very well when there is enough data; our mathematical model accounts for all of our novel empirical observations.

---

> > > > ### Comment · Reviewer_2Uvx · 2024-11-24
> > > >
> > > > Dear authors,
> > > >
> > > >   Thanks for your further explanation. However, I plan to keep my score. Thanks for your time!
> > > >
> > > > Reviewer 2Uvx

---

### Official Review · Reviewer_zJMu · 2024-11-04

**Soundness:** 1
**Presentation:** 1
**Contribution:** 2
**Rating:** 1
**Confidence:** 4

**Summary:**

The paper investigates Transformer behavior when trained from scratch to perform linear regression. It examines out-of-distribution (OOD) generalization across various settings, such as different ranges and distributions of linear functions.

**Strengths:**

The paper conducts thorough experiments across various scales and settings, providing a comprehensive analysis of Transformer behavior.

**Weaknesses:**

1. The related work could benefit from a more comprehensive review. The paper primarily discusses the works of Garg et al., Akyürek et al., and Von Oswald et al. on regression for in-context learning (ICL), but there are additional relevant studies in this area that are not cited. A more thorough literature review, covering empirical and theoretical works on regression in ICL, would enhance the paper’s context. Checking recent citations in this line of research may help identify key studies to include.

2. The notation in Section 4 could be clarified, as some symbols are difficult to interpret. For example, it’s not immediately clear what $\sigma$ represents in the context of $f_{i, \sigma}$. Additional explanations could help improve readability.

3. The organization of the paper could be refined to improve the overall flow. At times, the presentation feels somewhat informal, with experiments presented in sequence without clear connections, motivations, or in-depth analyses. For instance, it would be helpful if the authors could clarify the rationale for studying models of different scales and discuss what insights are gained from these comparisons. Additionally, mixing experiments on different scales and distributions makes it challenging to understand the primary conclusions. This structure could make it clearer to the reader what the authors aim to convey.

In general, I appreciate that the authors highlight the out-of-distribution (OOD) generalization issue for Transformers trained on linear regression, as initially noted by Garg et al. However, the experimental findings in Section 4 could be more impactful with clearer motivations and discussions. The hypothesis regarding induction heads and their role in OOD performance is somewhat interesting, though it could be strengthened with supporting theoretical insights or experimental validations, such as through mechanical interpolation. Presenting this hypothesis with additional rigor could provide more substantial contributions to the community.

**Questions:**

1. While it seems intuitive that, for instance, "9L6AH" refers to a model with 9 layers and 6 attention heads, this notation is somewhat non-standard. It would be helpful if the authors could define this notation explicitly before using it. Many other notations in the paper also follow this informal style, though I haven’t listed each instance. It would be beneficial if the authors could standardize and define these terms clearly at the outset.

2. In line 193, could the authors clarify whether it is $D^T_F$ or $D^t_F$? There are also several other typos throughout the paper that I haven't enumerated. Clarifying these would improve overall readability and precision.

3. In line 190, it’s unclear why the authors mention that the coefficients are in the range $[-1, 1]$, as this differs from the $N(0, 1)$ distribution. Additionally, there is no supporting figure or result indicating that coefficients within $[-1, 1]$ lead to zero MSE error. Given that I generally observe non-zero but small MSE error, it would be helpful if the authors could clarify this paragraph, particularly regarding the model size required to achieve zero average MSE error.

---

> ### Author Response · Authors · 2024-11-13
>
> We thank the reviewer for the comments.
>
> With regards to a more thorough lit review, we will add references to the revised version.  After submission to ICLR, we have already added related work not cited in the submitted version : Fu et al. (2023), Xie et al. (2021), Wu et al.(2023), Zhang et al. (2023), Panwar et al. (2023), Bai et al. (2024).  We would be glad to have more concrete suggestions about what to add over and above what the other reviewers have suggested.
>
> We have posted a new version of the paper with an improved organization and hopefully this will answer your criticism
>
> With regards to notation, we will clarify f_{i,\sigma}.  What it means is this: f_{i,\sigma} is the ith function sampled from N(0,\sigma] for 1 \leq \sgma \leq 10. We will clarify this notation.  We apologize for D^T_F which is a typo, D^t_F is the test distribution for the functions.  D^t_I is the distribution for the points x_i in the sequences to which the functions are applied; D_I is the distribution for the training sequences.  We will state this more clearly in the final version.
>
> Clarify the rationale for studying models of different scales and discuss what insights are gained from these comparisons. We wanted to see which components in actual  transformer based models were responsible for ICL and we wanted to see whether ICL improved with scale.  This puts us apart from a lot of the literature that has tried to show that under certain assumptions transformers CAN do ICL.  We want to know what they DO in practice. This was why we tested a number of models, including Attention Only models and MLP only models.  We will improve our description of the results in Table 1 and in section 4.6 where we summarize our findings.
>
> The reason we speak of coefficients a, b \in [-1,1] for functions ax +b is that given N(0,1) the model has seen functions with those coefficients many times (~70% of its training data in D_F), and that makes a difference to the overall success of its algorithm.  For instance consider the plots in Figure 4 where the coefficients are large This discussion is at the end of the background section. This is regardless of the size of the model though the largest models tested have essentially 0 error and the small ones have an error of 0.1 (see our Table 1).

---

> > ### Comment · Reviewer_zJMu · 2024-11-25
> >
> > Thank you for revising the paper. Below are my comments:
> >
> > While the paper highlights that several works claim Transformers can solve linear regression, it is important to acknowledge that a line of work has already identified counterexamples and brought this issue to the community’s attention. I suggest that the authors discuss and credit these efforts in the introduction or related work section, rather than implying that this paper is the first to address this issue. For instance, the generalization error in Transformers has been recognized in Garg et al.’s work, and other studies have examined similar issues in out-of-distribution (OOD) settings. Giannou et al. explored boundary value limitations in OOD ranges, while Zhang et al. addressed context-length generalization challenges. These studies collectively show a gap between theoretical expectations and experimental results, indicating that Transformers cannot solve linear regression reliably except in very simple scenarios.
> >
> > Relevant references include:
> > - Liu, Jerry Weihong, et al. “Can Transformers Solve Least Squares to High Precision?.” ICML 2024 Workshop on In-Context Learning.
> > - Shen, Lingfeng, Aayush Mishra, and Daniel Khashabi. “Do pretrained Transformers Really Learn In-context by Gradient Descent?.” arXiv preprint arXiv:2310.08540 (2023).
> > - Giannou, Angeliki, et al. “How Well Can Transformers Emulate In-context Newton’s Method?.” arXiv preprint arXiv:2403.03183 (2024).
> > - Zhang, Ruiqi, Spencer Frei, and Peter L. Bartlett. “Trained transformers learn linear models in-context.” arXiv preprint arXiv:2306.09927 (2023).
> >
> > The contribution of this work appears limited in scope. As noted above, the issue of Transformers failing to generalize has already been studied from various perspectives, such as sparsity in trained model weights and error precision in learned representations. While this paper adds evidence that Transformers struggle with strictly increasing or decreasing linear regression problems and boundary value issues, it primarily offers a hypothesis without substantial supporting evidence, whether theoretical or through mechanical interpolation. Expanding the work with stronger evidence or a deeper theoretical framework could significantly strengthen the contribution.

---

> > > ### Author Response · Authors · 2024-11-25
> > > **About Zhang et al.**
> > >
> > > Thank you for your clarification.   We did try to distinguish ourselves from Zhang et al 2024 or [1] in comments we made to another reviewer.  Here is what we said.
> > >
> > > Though similar to our work, [1] works with linear attention, whereas we look at attention layers as they actually are used with softmax (thus one could criticize [1] in the way you do in point 1 more than us at least on this score). In addition, [1] uses a new kind of optimization or training with gradients and a special fixed initial point. This means that their architecture and training are quite different from what normally happens with transformers; they are interested in getting a revised transformer-like model to learn linear functions, while we want to find out whether transformers as they actually are learn linear functions or something else. The results for the two architectures are quite different: While [1] says task shift does not affect their models, our task shifts affect the results in an important way, where we take D^{train}_H = N(0,1) (Our D_F) but D^{test}_H = N(0, \sigma) (our D^test_H) for 1 \leq \sigma \leq 10. Figure 1 clearly shows that for transformer models with soft attention, this task shift reduces performance dramatically. We also note unlike [1] that prompts that are too long induce chaotic behavior.
> > >
> > > In the covariate shift [1] also does something different from what we do. In covariate shift in [1], the distribution in the prompt is shifted but the distribution of the query stays the same. We do something different. When we take a distribution over input points in train D_I and set D^test_I \neq D_I, our shift is not the same; we shift both prompt and query distributions. With covariate shifts we found that the choice of points is important and model performance degrades considerably when the values of the functions on the chosen points lie beyond what we call boundary values. As far as we know we are the first to point out these boundary values and their dependence on model parameters. At least [1] does not do this. Our mathematical formulation of what models do, explains their behavior.
> > >
> > > You are right that we did not put all of this in the paper, and we should have a longer discussion.  We will do this in a revised version.

---

> ### Author Response · Authors · 2024-11-25
> **about Giannou et al.**
>
> We also want to clarify our relation to Giannou et al.  We cited and briefly discussed Giannou, Angeliki, et al. "How Well Can Transformers Emulate In-context Newton's Method?” in the revised version on the site and note that they have something like boundary values.  But their set up like Zhang et al's is also quite different from ours.
>
> There are three important differences.  First,  Giannou et al use linear self attention, which makes sense for the theoretical results, since they want to show that LLMs can in principle approximate Newton’s method. In this, they are like many other papers we cite, seeking to explain a model behavior by a theoretical construction. Our undertaking is different. We start from actual transformer architectures with soft attention and try to determine what they are actually doing in this task. We show that our models don’t do any of the reconstructions that we have seen proposed in the literature.
>
> Second, Giannou et al. also only examine differences in sampling the sequences of points in the prompt; i.e. they look at in our notation D_I \neq D^t_I, where D_I is the training distribution of points and D^t_I is the test distribution. In particular they fix a particular function and then input into it points that are in the training distribution and points outside of distribution. In distribution is defined as having been very likely to have been seen in training.  This is a special case of what we do, since we also shift the test distiribution for the functions sampled.  When we looked at cases where D_I \neq D^t_I, we saw that sometimes for input values within distribution, we got bad results (see figure 4 for example) when the function’s values on those points were outside what we call boundary values for all models tested, whereas Giannou et al. did not find any effects for models with 4 and more layers
> Third, to test Giannou et al use only very small models 4 ah 64 embeddings and go only to 6 layers, we go up to 12L8AH and d_emb = 256. They observe that after four layers, there is no significant improvement in performance by adding more layers. Those results are different from what we find (maybe due to their use of linear attention). We have included some comments on Giannou et al in a new revised version of the paper that we have submitted on the site, which we hope will reply to your concerns.

---

> ### Author Response · Authors · 2024-11-26
> **edited comment on Giannou et al. and a new version of the paper with comments on Zhang et al and Giannou et al.**
>
> Please find above a revised version of our comments on Giannou et al. and also see a revised version of the submission with comments included..  The new version defends our positive proposal for what the models are doing in more depth.  We hope to have addressed your concerns.

---

### Official Review · Reviewer_JBsD · 2024-11-06

**Soundness:** 2
**Presentation:** 2
**Contribution:** 2
**Rating:** 5
**Confidence:** 4

**Summary:**

This paper studies experimentally the setting of in-context learning linear regression . The authors reproduce the experiments of Garg et al and at inference time test the models with 1) different distributions for the input/weight vectors 2) larger values for the input/weight vectors.
Based on the observations of these results the authors argue that these models do not learn some type of algorithm.

**Strengths:**

Understanding what these models learn even in the setting of linear regression can significantly enhance our understanding of their capabilities limitations. Indeed it has been observed that the models do not generalize in out-of-distribution samples and thus it is unclear whether these models learn some type of algorithm.

**Weaknesses:**

1. The provided experimental study does not explain what these models are actually learning. For example it can be the case that the model are learning a tailor-made preconditioned gradient descent type of algorithm, with the preconditioned matrix being optimal for the in-distribution values and sub-optimal for out-of-distribution values.
2. It cannot be excluded that the current training methods are not optimal, since we know that these models do have the capability of representing these algorithms.
3. Some of these results have already been observed experimentally for example see [1] (Figures 5,6).  In these experiments consider multi-dimensional linear regression, they keep all expect for one dimension fixed and plot how the function changes when varying one dimension from [-B,B] similar to the authors' experiments for one dimensional linear regression.

In general the main weakness of this paper is that it does not make a convincing argument towards what these models are actually learning.
[1]: Giannou, Angeliki, et al. "How Well Can Transformers Emulate In-context Newton's Method?." arXiv preprint arXiv:2403.03183 (2024).

**Questions:**

I agree with the authors that these models do not exactly learn some type of algorithm but I think that the main question is why these models do not do so while they have the expressivity ? One possible explanation is that there exist  parameters that better interpolate the specific distributions, while existing algorithms work for any type of distribution.

Did the authors try to train the models with multiple distributions? It could be the case that then the models are able to perform some type of algorithm by not fine-tuning their weights to fit a specific distribution. Furthermore, considering the second point above, did the authors perform a search over the hyperparameters for training?

---

> ### Author Response · Authors · 2024-11-13
>
> We thank the reviewer for the remarks and questions.
>
> We agree with the reviewer that it is an interesting question why the models don’t do what they are capable of.  But we were interested in a different, equally interesting (to us) question:  what are the models actually doing in this very simple task?  First we showed that they are not learning an algorithm like least squares.  In the last section of the paper, we provide a mathematical model of what they are doing; they are treating the sequences not as the graph of a function but as just a sequence.  They develop an algorithm using something like Olsson et al’s induction heads to interpolate a next value for the given sequence in the prompt from close-by sequences in their pre-training.
>
> Did we try to train with multiple distributions? Yes in section 4 we look at three different distributions for function sampling: N, bimodal and uniform (section 4.2).
>
> We do say what the models are actually learning, see section 5 of paper.  We will clarify that section in our revised version. We have highlighted it also in the abstract.
>
> We will clarify that current training methods are not optimal. We wanted to investigate training and prompting methods already in the literature; we show that the proposed methods don't do what is claimed.
>
> We will certainly cite and discuss Giannou, Angeliki, et al. "How Well Can Transformers Emulate In-context Newton's Method?”, an interesting paper, in our revised version.  It uses linear self attention, which makes sense for the theoretical results, since they want to show that LLMs can in principle approximate Newton’s method.  In this, they are like many other papers we cite, seeking to explain a model behavior by a theoretical construction.  Our undertaking is different.  We start from actual transformer architectures with soft attention and try to determine what they are actually doing in this task.  We in fact show that they don’t do any of the reconstructions that we have seen proposed in the literature.
> Giannou et al. also only examine differences in sampling the sequences of points in the prompt; i.e. they look at in our notation  D_I \neq D^t_I, where D_I is the training distribution of points and D^t_I is the test distribution.  In particular they fix a particular function and then input into it points that are in the training distribution and points outside of distribution.  In distribution is defined as having been very likely to have been seen in training.  However, when we have looked at cases where D_I \neq D^t_I, we saw that sometimes for input values within distribution, we got bad results  (see figure 4 for example) when the function’s values on those points were outside what we call boundary values.
> In addition to examining  D_I \neq D^t_I, we also vary the distributions from which we sample functions, D_F and D^t_F–i.e. values of the points chosen.  Giannou et al did not do this.
>
> Finally,  to test Giannou et al use only very small models 4 ah 64 embeddings and go only to 6 layers, we go up to 12L8AH and d_emb = 256.
> They observe that after four layers, there is no significant improvement in performance by adding more layers. Those results are different from what we find (maybe due to their use of linear attention).  We have included some comments on Giannou et al and clarified how we differ from the related work we know about in a revised version of the paper that we have submitted on the site.

---

### Note · Authors · 2024-11-28

I have read and agree with the venue's withdrawal policy on behalf of myself and my co-authors.